# Pattern and prevalence of vaping nicotine and non-nicotine drugs in the United Kingdom: a cross-sectional study

Emmert Roberts ,[1,2] Eve Taylor,[1] Sharon Cox,[3] Leonie Brose ,[1] Ann McNeill ,[1] Deborah Robson[1]

¹National Addiction Centre, Institute of Psychiatry, Psychology and Neuroscience, King's College London, London, UK
²South London and the Maudsley NHS Foundation Trust, London, UK
³Institute of Epidemiology & Health, Faculty of Population Health Sciences, University College London, London, UK

**Correspondence to**
Dr Deborah Robson;
deborah.j.robson@kcl.ac.uk

## ABSTRACT

**Objectives** Electronic vaping devices are being used to consume nicotine and non-nicotine psychoactive drugs. We aimed to determine the pattern and prevalence of using vaping devices for nicotine and/or non-nicotine drug administration in the United Kingdom and how these differ by drug type and individual sociodemographic characteristics. We explored reasons for vaping onset and continuation.

**Design** An online cross-sectional survey

**Participants** A convenience sample of adults (aged ≥18 years) in the UK.

**Primary and secondary outcome measures** The primary outcome was prevalence of current use (within the last 30 days) of a vaping device to administer either nicotine or 18 types of non-nicotine drugs. We additionally evaluated reasons for onset and continuation of vaping. Sociodemographic characteristics were compared between the UK general population using census data and those vaping non-nicotine drugs.

**Results** We recruited 4027 participants of whom 1637 (40.7%) had ever used an electronic vaping device; 1495 (37.1%) had ever vaped nicotine and 593 (14.7%) had ever vaped a non-nicotine drug. Overall, 574 (14.3%) currently vaped nicotine and 74 (1.8%) currently vaped a non-nicotine drug. The most common currently vaped non-nicotine drug was cannabis (n=58, 1.4%). For nicotine, people's modal reasons to start and continue vaping was to quit smoking tobacco. For almost all other drugs, people's modal reason to start vaping was curiosity and to continue was enjoyment. Compared with the general population, the population who had ever vaped a non-nicotine drug were significantly younger, had more disabilities and fewer identified as white, female, heterosexual or religious.

**Conclusions** A non-trivial number of people report current use and ever use of an electronic vaping device for non-nicotine drug administration. As vaping technology advances and drug consumption changes, understanding patterns of use and associated behaviours are likely to be increasingly important to both users and healthcare professionals.

## STRENGTHS AND LIMITATIONS OF THIS STUDY

⇒ The study reports, to our knowledge, the largest cross-sectional adult sample of the sociodemographic profile of people using electronic vaping devices to consume non-nicotine drugs, alongside people's reasons for onset and continuation of vaping and the type of devices they use.

⇒ The study relied on self-report to identify non-nicotine drug use and may be subject to social desirability bias or participant misinterpretation.

⇒ The study was conducted using an online convenience sample and several populations who have historically higher rates of non-nicotine drug consumption may have been excluded due to its digital nature or other life circumstances (eg, people experiencing homelessness or currently serving a prison sentence).

## INTRODUCTION

In the last decade the advent of electronic nicotine vaping products has changed how some people use nicotine and recent reports also confirm use of electronic vaping devices to administer non-nicotine psychoactive drugs.[1–4] As innovation in technology develops, the prevalence and patterns of using electronic vaping devices to facilitate non-nicotine drug use are subject to change.[5] While in some circumstances vaping non-nicotine drugs may present fewer harms when compared with other routes of administration (eg, smoking), in some cases vaping could potentially pose unique harms due to the speed of onset of drug effects, alongside the concomitant use of any adulterants.[5–7] Most notably the 2019 US outbreak of acute lung injury associated with vaping illicit cannabinoid products contaminated with vitamin E acetate highlights the potential for associated harms.[8]

Monitoring of nicotine vaping in the United Kingdom is well documented,[9] while data on vaping of non-nicotine drugs are sparse.[10–12] Understanding the prevalence and patterns of vaping non-nicotine drugs is important for healthcare professionals to both facilitate discussions regarding consumers' preferred routes of drug administration, and to provide

appropriate harm reduction advice. While a recent review of online drug user forums has identified over 300 reports related to the manufacture and/or use of e-liquids containing non-nicotine drugs,[2] few epidemiological studies have been conducted to determine the prevalence and patterns of using electronic vaping devices to administer different non-nicotine drugs. Although some studies have examined vaping in the context of nicotine and cannabis administration,[10–12] to our knowledge, only a single cross-sectional study conducted in 2017 in adults in the UK reported that 340 (13.6%) participants had ever used an electronic vaping device for any type of recreational drug administration during their lifetime.[3] However, this estimate has not been reproduced, the sociodemographic characteristics of the cohort of people vaping non-nicotine drugs have not been fully evaluated and there has been little examination of why people use vaping as a route of non-nicotine drug administration.

To address this gap, we aimed to determine the pattern and prevalence of using electronic vaping devices for nicotine and non-nicotine drug administration in the UK and how these differ by drug type and individual sociodemographic characteristics. In addition, we aimed to examine individual reasons for the onset of vaping non-nicotine drugs and why people continue vaping as a route of administration.

## METHODS

An anonymous voluntary cross-sectional survey recruiting a convenience sample of adult participants (aged ≥18 years) through the online Prolific platform.[13] Prolific, using their survey panel of research respondents based in the UK, advertised the study to all adults on their platform.[14] The study protocol, questionnaire and statistical analysis plan were preregistered on the Open Science Framework and are available at https://osf.io/bdfhp/. The survey went live on 10 March 2022 with the prespecified minimum sample size of 3275 being reached within 24 hours.

The survey used sociodemographic questions identical to those within the UK census and questions relating to vaping were based on the previous survey by Blundell *et al* and adapted by the research team to include questions relating to additional substances not previously captured, vaping device type and reasons behind vaping.[3 15] The final published questionnaire is additionally available in the online supplemental material. Participants were required only to complete sections of the questionnaire applicable to their self-described vaping status of each substance and answers forced prior to moving onto the subsequent section thus no missing data could be generated.

The primary outcome was the prevalence of current use of a vaping device to administer either nicotine or 18 specific types of non-nicotine drugs, defined as use in the past 30 days (a complete list of all non-nicotine drug types and subtypes studied can be found in the online supplemental table S1). Secondary outcomes included

prevalence of ever using a vaping device to administer either nicotine or non-nicotine drugs and current use of nicotine or non-nicotine drugs by routes of administration other than vaping. We additionally collected if participants considered any currently consumed drug was having a negative consequence on their mental or physical health (harmful use) and on frequency of current use and vaping, age of first use and vaping onset, reasons for onset of vaping and why people continue vaping as a route of administration.

Individual UK census defined sociodemographic characteristics were also collected including level of educational attainment and eight 'protected' characteristics as defined by the UK Equality Act (age, sex, disability, gender identity, sexual orientation, ethnicity, religion and marital status).[16]

Sociodemographic and vaping characteristics were compared between those who had used an electronic vaping device to vape a non-nicotine drug and those who had never used an electronic vaping device. Additionally, study sample sociodemographic characteristics were compared with the UK general population using publicly available data from the UK census.[17] Continuous variables were compared using an unpaired t-test, and categorical variables using Fisher's exact test, all comparisons were bivariate. Where there were more than two levels to a categorical variable, adjusted residuals, where the significance level of $\alpha=0.01$, were calculated to identify which individual cells contributed to between-group differences.[18] All analyses were conducted using STATA SE V.15.1 with the significance level set at 0.05. The study was reported in accordance with the STrengthening the Reporting of OBservational studies in Epidemiology statement for observational studies.[19]

## PATIENT AND PUBLIC INVOLVEMENT
No patient or public involvement.

## RESULTS
There were a total of 4027 participants of whom 1637 (40.7%) had ever used an electronic vaping device; 1495 (37.1%) people had ever vaped nicotine and 593 (14.7%) had ever vaped a non-nicotine drug. A total of 1044 (25.9%) had only ever exclusively vaped nicotine, 142 (3.5%) had only ever exclusively vaped a non-nicotine drug and 451 (11.2%) had ever vaped both nicotine and a non-nicotine drug. The majority of the overall sample were aged between 25 and 44 (57.9%), predominantly identified as female (64.9%), and predominantly identified as being from a white ethnicity (80.8%) (table 1).

About a fifth (n=834, 20.7%) were currently using a nicotine product; 595 (14.8%) were currently smoking tobacco, 574 (14.3%) were currently vaping nicotine and 280 (7.0%) were currently concurrently smoking tobacco and vaping nicotine. Fewer (n=217, 5.4%) participants were currently using a non-nicotine drug, of whom 74

**Table 1** The age, sex and ethnicity breakdown of those participants who have never vaped, ever vaped any drug including nicotine and ever vaped any non-nicotine drug

| Sociodemographic characteristic | | All n (%) | Never vaped n (%) | Ever vaped any drug (including nicotine) n (%) | Ever vaped any non-nicotine drug n (%) | Ever vaped any non-nicotine drug compared with never vaped (p)* | Ever vaped any non-nicotine drug compared with UK general population (p)* |
|---|---|---|---|---|---|---|---|
| All | All | 4027 (100%) | 2390 (100%) | 1637 (100%) | 593 (100%) | – | – |
| Age | 18–24 | 441 (11.0%) | 199 (8.3%) | 242 (14.8%) | 112 (18.9%) | <0.001† | <0.001† |
| | 25–34 | 1265 (31.4%) | 670 (28.0%) | 595 (36.4%) | 244 (41.2%) | | |
| | 35–44 | 1067 (26.5%) | 632 (26.4%) | 435 (26.6%) | 146 (24.6%) | | |
| | 45–54 | 629 (15.6%) | 410 (17.2%) | 219 (13.4%) | 57 (9.6%) | | |
| | 55–64 | 428 (10.6%) | 319 (12.4%) | 109 (6.7%) | 25 (4.2%) | | |
| | 65–74 | 171 (4.3%) | 138 (5.8%) | 33 (2.0%) | 8 (1.4%) | | |
| | Over 75 | 26 (0.7%) | 22 (0.9%) | 4 (0.2%) | 1 (0.2%) | | |
| Sex | Male | 1400 (34.8%) | 754 (31.2%) | 655 (40.1%) | 271 (45.7%) | <0.001‡ | <0.001‡ |
| | Female | 2614 (64.9%) | 1641 (68.7%) | 973 (59.4%) | 317 (53.5%) | | |
| | Do not wish to say | 13 (0.3%) | 4 (0.2%) | 9 (0.6%) | 5 (0.8%) | | |
| Ethnicity | English/Welsh/Scottish/Northern Irish/British | 3254 (80.8%) | 1952 (81.7%) | 1302 (79.5%) | 437 (73.7%) | <0.001§ | <0.001§ |
| | Irish | 42 (1.0%) | 27 (1.1%) | 15 (0.9%) | 9 (1.5%) | | |
| | Gypsy or Irish traveller | 2 (0.1%) | 0 (0.0%) | 2 (0.1%) | 1 (0.2%) | | |
| | Any other white background | 227 (5.6%) | 128 (5.4%) | 99 (6.1%) | 42 (7.1%) | | |
| | White and black Caribbean | 37 (0.9%) | 17 (0.7%) | 20 (1.2%) | 7 (1.2%) | | |
| | White and black African | 14 (0.4%) | 8 (0.3%) | 6 (0.4%) | 4 (0.7%) | | |
| | White and Asian | 38 (0.9%) | 14 (0.6%) | 24 (1.5%) | 15 (2.5%) | | |
| | Any other mixed/multiple ethnic background | 21 (0.5%) | 12 (0.5%) | 9 (0.6%) | 4 (0.7%) | | |
| | Indian | 78 (1.9%) | 42 (1.7%) | 36 (2.2%) | 16 (2.7%) | | |
| | Pakistani | 52 (1.3%) | 27 (1.1%) | 25 (1.5%) | 11 (1.9%) | | |
| | Bangladeshi | 17 (0.4%) | 12 (0.5%) | 5 (0.3%) | 2 (0.3%) | | |
| | Chinese | 58 (1.4%) | 37 (1.6%) | 21 (1.3%) | 12 (2.0%) | | |
| | Any other Asian background | 24 (0.6%) | 13 (0.5%) | 11 (0.7%) | 5 (0.8%) | | |
| | African | 69 (1.7%) | 52 (2.2%) | 17 (1.0%) | 7 (1.2%) | | |
| | Caribbean | 29 (0.7%) | 18 (0.8%) | 11 (0.7%) | 3 (0.5%) | | |
| | Any other Black/African/ Caribbean background | 5 (0.1%) | 1 (0.1%) | 4 (0.2%) | 2 (0.3%) | | |
| | Arab | 14 (0.4%) | 7 (0.3%) | 7 (0.4%) | 4 (0.7%) | | |
| | Any other ethnic group | 21 (0.5%) | 8 (0.3%) | 13 (0.8%) | 7 (1.2%) | | |
| | Do not wish to say | 25 (0.6%) | 15 (0.6%) | 10 (0.6%) | 5 (0.8%) | | |

*Compared using Fishers exact test.
†For the purpose of statistical testing this category was collapsed into four levels: 18–24, 25–34, 35–44 and over 45.
‡For the purpose of statistical testing this category was collapsed into two levels: male and female.
§For the purpose of statistical testing this category was collapsed into two levels: white (English/Welsh/Scottish/Northern Irish/British, Irish, Gypsy or Irish Traveller or any other white background) and non-white (white and black Caribbean, white and black African, white and Asian, any other mixed/multiple ethnic background, Indian, Pakistani, Bangladeshi, Chinese, any other Asian background, African, Caribbean, any other black/African/Caribbean background, Arab or any other ethnic group).

(1.8%) were currently vaping a non-nicotine drug. The only non-nicotine drugs currently being vaped were cannabis 58 (1.4%), caffeine 12 (0.3%), alcohol 5 (0.1%), opioids 1 (0.02%) and benzodiazepines 1 (0.02%).

A profile of current use for those drugs currently being vaped is available in table 2 with a breakdown for all other drugs available in the online supplemental table S2. Participants' mean age at first use by any route was statistically significantly lower than their mean age at first vaping for nicotine, alcohol, caffeine, cannabis, synthetic cannabinoids and cocaine. For nicotine, people's modal reason to try vaping and their current reason to vape was to quit smoking tobacco. For almost all other drugs, people's modal reason to try vaping was curiosity and their modal reason to currently vape was because of enjoyment. The modal type of electronic vaping device currently used was a commercially bought electronic or e-cigarette for all studied drugs except cannabis for which the model type of device currently used was a commercially bought electronic device to vaporise dry herbs.

Of the 593 people who had ever vaped a non-nicotine drug, 484 (12.0% of the overall sample) had only ever vaped one type of non-nicotine drug and 109 (2.7%) had vaped two or more types of non-nicotine drugs. The most frequently ever vaped non-nicotine drug was cannabis (n=454, 11.3%), of which 129 (3.2%) people had only ever vaped cannabidiol (CBD) products. This was followed by caffeine (n=96, 2.4%), alcohol (n=53, 1.3%), synthetic cannabinoids (n=45, 1.1%), dimethyltryptamine (n=16, 0.4%), cocaine (n=10, 0.2%), psilocybin (n=7, 0.2%), ketamine (n=7, 0.2%), opioids (n=6, 0.1%), MDMA (n=6, 0.1%), 2Cs (n=6, 0.1%), amphetamines (n=5, 0.1%), gamma-butyrolactone (n=4, 0.1%), benzodiazepines (n=4, 0.1%), mephedrone (n=3, 0.1%), alpha-PVP (n=2, 0.1%) and NBOMe (n=1, 0.02%). A bar chart depicting the prevalence of ever and current vaping for each drug is available in the online supplemental figure S1.

When comparing the study participants who had ever vaped a non-nicotine drug to both the UK general population and to those study participants who had never vaped, they were significantly different in every studied protected characteristic sociodemographic category. This group was significantly younger (with significantly more people aged <35 and significantly fewer aged >45), had a higher proportion of males, there were fewer people identifying with white ethnicities, there were more individuals who identified as LGBT+, more non-religious individuals, more people with disabilities and fewer married people. However, there was no statistically significant difference by level of educational attainment. A breakdown by age, sex and ethnicity can be found in table 1, with the remaining sociodemographic differences available in the online supplemental table S3.

## DISCUSSION

In this study of 4027 respondents, 1.8% reported they were currently vaping a non-nicotine drug and 14.7% reported ever having vaped a non-nicotine drug. About one in six respondents were currently vaping nicotine around half of whom were also smoking tobacco.

Current prevalence of smoking tobacco among participants in our study (14.8%) was similar to those from other surveys in 2022 in England, such as estimates from the Smoking Toolkit Study (STS, 15.0%) and the Action on Smoking and Health (ASH) adult survey (13.2%).[20 21] However, our estimate of current vaping was higher, at 14.5%, than estimates from the STS survey (9.3%) and ASH adult survey (8.3%).[20 21] This is possibly due to sampling differences or how prevalence was estimated, with our survey including all adults who have vaped any substance in the past 30 days as current vapers, therefore likely capturing occasional/social/one-off use among people who would not identify themselves as 'vapers'. Additionally, the day prior to data collection was national 'No Smoking Day' in the UK which may have affected estimates.

The prevalence of ever using a vaping device to administer a non-nicotine drug was similar to estimates from a previous study (14.7% vs 13.6%),[3] however prevalence estimates of currently vaping a non-nicotine drug were substantially lower (1.8% vs 9.4%).[3 10] This could be explained by sampling differences or may perhaps demonstrate that current prevalence may fluctuate depending on technological or other changes in the vaping device market with new products potentially leading to curiosity-driven experimentation that may not be sustained over time.[22] Temporal prevalence trends are likely to reveal more about these relationships than isolated cross-sectional sampling.

When compared with the general population, the non-nicotine drug vaping population were significantly different across all studied protected characteristics, however their sociodemographic profile was broadly comparable to those of non-nicotine drug users in publicly available UK cross-sectional data, for example, the Crime Survey for England and Wales.[23] Participants' mean age at first use by any route was consistently lower than respondents mean age at first vaping for all studied drugs, although lack of statistical power may account for the fact this was only significant for a few substances. This suggests that vaping is generally a rare first route of drug administration.

Similar to other studies, cannabis was the most commonly vaped non-nicotine drug with 11.3% of total respondents having ever vaped a cannabis product, however just over a quarter of respondents had only ever vaped a product containing CBD which, at the time of the survey, was legal to purchase and consume in the UK.[3 24] This underscores vaping should routinely be enquired about as a potential method of administration among cannabis users, and any future work on understanding relative cannabis-related harms should include vaping when considering routes of administration. Concerningly 45 individuals reported vaping synthetic cannabinoids, which have been linked to two vaping product-related deaths in the UK.[7]

**Table 2** Current use profiles for all drugs which the participants were currently vaping (within the last 30 days)

| | Any (including nicotine) | Any non-nicotine drug | Nicotine | Caffeine | Alcohol | Cannabis | Opioids | Benzodiazepines |
|---|---|---|---|---|---|---|---|---|
| Current use, n (%) | 929 (100%) | 217 (100%) | 834 (100%) | 45 (100%) | 29 (100%) | 146 (100%) | 2 (100%) | 1 (100%) |
| Frequency of current use, n (%) | | | | | | | | |
| Daily | – | – | 601 (72.1%) | 28 (62.2%) | 4 (13.8%) | 58 (39.7%) | 1 (50.0%) | 0 (0.0%) |
| Weekly | – | – | 118 (14.2%) | 16 (35.6%) | 14 (48.3%) | 45 (30.8%) | 1 (50.0%) | 0 (0.0%) |
| Monthly | – | – | 115 (13.8%) | 1 (2.2%) | 11 (37.9%) | 43 (29.5%) | 0 (0.0%) | 1 (100.0%) |
| Currently self-report harmful use, n (%) | 542 (58.3%) | 73 (33.6%) | 508 (60.9%) | 14 (31.1%) | 13 (44.8%) | 47 (32.2%) | 1 (50.0%) | 0 (0.0%) |
| Currently vape, n (%) | 621 (68.8%) | 74 (34.1%) | 574 (68.8%) | 12 (26.7%) | 5 (17.2%) | 58 (39.7%) | 1 (50.0%) | 1 (100.0%) |
| Frequency of current vaping, n (%) | | | | | | | | |
| Daily | – | – | 363 (43.5%) | 7 (15.6%) | 0 (0.0%) | 14 (9.6%) | 0 (0.0%) | 0 (0.0%) |
| Weekly | – | – | 130 (15.6%) | 4 (8.9%) | 2 (6.9%) | 22 (15.1%) | 1 (50.0%) | 1 (100.0%) |
| Monthly | – | – | 81 (9.7%) | 1 (2.2%) | 3 (10.3%) | 22 (15.1%) | 0 (0.0%) | 0 (0.0%) |
| Age first used mean (SE) | 17.2 (0.15) | 19.8 (0.36) | 17.1 (0.15) | 22.3 (1.24) | 18.3 (1.09) | 19.1 (0.34) | 24.2 (5.40) | 25.0 (4.56) |
| Age first vaped mean (SE) | 29.8 (0.28) | 27.9 (0.42) | 30.2 (0.30) | 27.6 (1.83) | 24.6 (1.4) | 28.1 (0.46) | 35.3 (5.90) | 29.8 (7.86) |
| t, p* | –41.5, <0.001 | –20.3, <0.001 | –41.2, <0.001 | –6.8, <0.001 | –4.3, <0.001 | –19.7, <0.001 | –2.2, 0.08 | –1.3, 0.27 |
| Modal reason for vaping initially | – | – | To quit smoking tobacco (31.7%) | Curiosity just wanted to try (37.5%) | Curiosity just wanted to try (50.9%) | Curiosity just wanted to try (41.6%) | Curiosity just wanted to try (50.0%) | Curiosity just wanted to try (25.0%) |
| Modal reason for vaping currently | – | – | To quit smoking tobacco (23.0%) | Because I enjoy it (41.7%) | Because I enjoy it (60.0%) | Because I enjoy it (34.5%) | For some other reason (100.0%) | Because I enjoy it (100.0%) |
| Stated an intention to quit, n (%) | 567 (61.0%) | 53 (24.4%) | 541 (64.9%) | 14 (31.1%) | 9 (31.0%) | 29 (19.9%) | 1 (50.0%) | 1 (100%) |
| Modal type of electronic vaping device currently used (%) | – | – | Commercially bought e-cigarette (86.4%) | Commercially bought e-cigarette (75.0%) | Commercially bought e-cigarette (40.0%) | Commercially bought electronic device used to vaporise dry herbs (53.5%) | Commercially bought e-cigarette (100.0%) | Commercially bought e-cigarette (100.0%) |

*Compared using an unpaired t-test.

While the study is, to our knowledge, the largest cross-sectional sample in the UK published literature of people ever vaping a range of non-nicotine drugs there are several limitations. The study relied on self-report to identify drug use and may therefore be subject to social desirability bias and was conducted using an online convenience sample. Several populations who have historically higher rates of non-nicotine drug consumption may have been excluded due to the study's digital nature or other life circumstances (eg, people experiencing homelessness or currently serving a prison sentence). This is particularly important given the harms associated with drug use are disproportionately higher within these populations.[7] The small numbers of people vaping specific substances prevent robust statistical analyses due to small cell sizes and as such statistical testing should be interpreted with caution. Future studies with larger sample sizes or meta-analyses may be able to overcome these limitations. Additionally, despite the survey questions stating that flavoured e-liquids such as coffee and alcohol flavours do not contain either caffeine or alcohol, we cannot be sure that respondents did not misinterpret these questions and thus reported vaping prevalence results for alcohol and caffeine may represent overestimates.

Understanding people's experiences and perceptions of vaping as a means of administering non-nicotine drugs is likely to be increasingly important both to users and to healthcare and policy professionals. This study demonstrates that a non-trivial number of people have ever consumed non-nicotine drugs using vaping devices and understanding how this prevalence changes over time and which individuals may be most at risk of harm or benefit from this administration method require further exploration. Future research should be conducted in larger non-UK samples, examine any potential trends in use of vaping products to administer non-nicotine drugs and examine vaping in samples known to experience disproportionate drug-related harms.

**Contributors**  ER: conceptualisation, methodology, investigation, data curation, formal analysis, project administration, supervision and funding acquisition, writing original draft. ET: methodology, investigation, data curation, formal analysis, project administration, supervision, writing review and editing. SC: investigation, writing review and editing. LB: investigation, writing review and editing. AM: conceptualisation, supervision, writing review and editing. DR: conceptualisation, methodology, investigation, supervision, funding acquisition, writing review and editing. ER is the article guarantor and accepts full responsibility for the work and/or the conduct of the study, had access to the data, and controlled the decision to publish.

**Funding**  This study was supported by the National Institute of Health Research (NIHR) Biomedical Research Centre at South London and Maudsley NHS Foundation Trust and King's College London. ER is funded by a Commonwealth Fund Harkness Fellowship in Healthcare Policy and Practice. SC receives salary support from Cancer Research UK (PRCRPG-Nov21\100002). ET, DR and AM are funded by the National Institute for Health and Care Research (NIHR) Health Protection Research Unit in Environmental Exposures and Health, a partnership between the UK Health Security Agency and Imperial College London. DR and AM are also funded by the National Institute for Health Research (NIHR) Applied Research Collaboration (ARC) South London at King's College Hospital NHS Foundation Trust. AM, DR and LB are members of SPECTRUM, a UK Prevention Research Partnership Consortium (MR/S037519/1). UKPRP is an initiative funded by the UK Research and Innovation Councils, the Department of Health and Social Care (England) and the UK devolved administrations, and leading health research charities. The funders had no role in: study design, in the collection, analysis, or interpretation of data; in the writing of the report; or in the decision to submit the paper for publication. The views expressed in this article are those of the authors and not necessarily those of the Commonwealth Fund, NIHR, or the Department of Health and Social Care.The funders had no role in: study design, in the collection, analysis, or interpretation of data; in the writing of the report; or in the decision to submit the paper for publication. The views expressed in this article are those of the authors and not necessarily those of the Commonwealth Fund, NIHR, or the Department of Health and Social Care.

**Competing interests**  None declared.

**Patient and public involvement**  Patients and/or the public were not involved in the design, or conduct, or reporting or dissemination plans of this research.

**Patient consent for publication**  Not applicable.

**Ethics approval**  Informed consent was obtained from all participants and ethical approval was gained from the King's College London Health Faculties Research Ethics Subcommittee (Reference: HR-21/22-26249). Participants gave informed consent to participate in the study before taking part.

**Provenance and peer review**  Not commissioned; externally peer reviewed.

**Data availability statement**  Data are available upon reasonable request. Data are available from the corresponding author upon resonable request, subject to institutional and ethical approval.

**ORCID iDs**
Emmert Roberts http://orcid.org/0000-0002-4152-5570
Leonie Brose http://orcid.org/0000-0001-6503-6854
Ann McNeill http://orcid.org/0000-0002-6223-4000

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
