## [Reviewer comments · BMJ Open]

ARTICLE DETAILS

TITLE (PROVISIONAL)	The pattern and prevalence of vaping nicotine and non-nicotine drugs in the United Kingdom: A cross-sectional study
AUTHORS	Roberts , Emmert; Taylor, Eve; Cox, Sharon; Brose, Leonie; McNeill, Ann; Robson, Deborah

VERSION 1 – REVIEW

REVIEWER	Gomaa, Huda Alexandria University
REVIEW RETURNED	17-Aug-2022

GENERAL COMMENTS	The authors conducted an important cross-sectional survey among adults for increasingly hot topic of vaping. The paper missed highlighting the significance of their results in the context of findings from Smokefree GB survey and The Smoking Toolkit Study. The agreement and disagreement between surveys that discuss the same health issue of vaping will add value to readers.
--

REVIEWER	Terry-McElrath, Yvonne University of Michigan, Institute for Social Research
REVIEW RETURNED	15-Dec-2022

GENERAL COMMENTS	The authors have undertaken a timely and important issue, that of vaping non-nicotine substances. They have utilized a convenience sample (oddly only one day of data collection) to collect data on vaping prevalence and a very large number of additional measures. I am concerned that while the topic chosen is certainly important, the resulting manuscript has significant limitations that prevent me from recommending further review. Major comments 1. The abstract states two very large goals for the paper: a) reporting the “pattern and prevalence of using vaping devices” and how such behaviors “differ by drug type and individual sociodemographic characteristics” and b) exploring “reasons for vaping onset and continuation”. The resulting manuscript is, however, much too short to adequately address either goal.2. The abstract also states that sociodemographic and vaping characteristics will be compared with the UK general population. At no point do the authors actually explain how they are doing this comparison – they cite data from the UK census, but exactly how such data are compared with their own estimates is unknown. The authors conclude that “compared to the general population, the population who had ever vaped...” but this conclusion is not justified using the current data provided.3. The review of literature seems very inadequate. The authors state their study is the largest cross-sectional sample to examine these
--

	issues, and the only one to investigate sociodemographic associations. A growing body of research has investigated non-nicotine vaping, as well as reasons for vaping, and the authors have not recognized this body of work (see, for example, US studies by Keyes et al 2022; Miech et al 2019; Patrick et al., 2020; and UK studies by Sund et al https://doi.org/10.1093/qjmed/hcac220 and Fataar et al https://pubmed.ncbi.nlm.nih.gov/31731420/). Citing only Blundell et al. is questionable. 4. A great deal more information is needed regarding methods. What is the Prolific platform, and what are the strengths/limitations of its use? Why were data collected only on one day (March 10, 2022)? What hours? What were the limitations of this? How might it have affected results? The actual wording for measures used needs to be provided, not merely briefly described. How were missing data handled, and what was the extent of missing? Clarification of the analyses is also needed; were all models bivariate? 5. The analytic approach is lacking. Use of global chi square tests and t-tests does not provide the ability to identify where specific differences are found. 6. The authors are to be commended on their careful reporting in the tables, but these indicate that the sample sizes used are simply too small to provide reliable estimates for many groups. Cell sizes are frequently shown of less than 5. 7. No descriptive information is provided for the overall sample. Minor comments Abstract P. 4 line 8: recommend clarifying the meaning of non-nicotine drugs, as there are flavorings, etc. P. 4 line 16: provide age range Introduction P. 6, lines 9-10: use “confirm” and not “suggest”; this research is clear on this issue. Also, define non-nicotine drugs at first use
--	---

VERSION 1 – AUTHOR RESPONSE

Reviewer: 1
 Dr. Huda Gomaa, Alexandria University

Comments to the Author:

The authors conducted an important cross-sectional survey among adults for increasingly hot topic of vaping.

Thank you for your comments.

The paper missed highlighting the significance of their results in the context of findings from Smokefree GB survey and The Smoking Toolkit Study. The agreement and disagreement between surveys that discuss the same health issue of vaping will add value to readers.

Thank you for your comment. We agree with the reviewer that specifically contextualising our results alongside the findings of the Smokefree GB survey and Smoking Toolkit Study would add value. Neither of these studies currently collects data on non-nicotine vaping however we have expanded our text in the discussion section on page 13 on comparisons with national smoking and vaping prevalence estimates to highlight similarities and differences in findings “Our current prevalence of smoking tobacco (14.8%) was similar to those from other surveys in 2022 in England, such as estimates from the Smoking Toolkit Study (STS) (15.0%) and the Action on Smoking and Health (ASH) adult survey (13.2%). (18, 19) However, our estimate of current vaping was higher, at 14.5%,

than estimates from the STS survey (9.3%) and ASH-Adult survey (8.3%). (18, 19) This is possibly due to sampling differences or how prevalence was estimated, with our survey including all adults who have vaped any substance in the past 30 days as current vapers, therefore likely capturing occasional/social/one-off use.”

Reviewer: 2

Dr. Yvonne Terry-McElrath, University of Michigan

Comments to the Author:

The authors have undertaken a timely and important issue, that of vaping non-nicotine substances. They have utilized a convenience sample (oddly only one day of data collection) to collect data on vaping prevalence and a very large number of additional measures. I am concerned that while the topic chosen is certainly important, the resulting manuscript has significant limitations that prevent me from recommending further review.

Thank you for your comments. We have clarified the reason data collection occurred only on a single day and added to the methods section on page 6. Our pre-specified sample size calculation, available as part of the pre-registered protocol and statistical analysis plan on the Open Science Framework (<https://osf.io/bdfhp/>), suggested the minimum sample size required was 3,275. This was surpassed within 24 hours and as such data collection occurred over a single twenty-four period. “The survey went live on March 10th 2022 with the pre-specified minimum sample size of 3,275 being reached within twenty-four hours”

Major comments

1. The abstract states two very large goals for the paper: a) reporting the “pattern and prevalence of using vaping devices” and how such behaviors “differ by drug type and individual sociodemographic characteristics” and b) exploring “reasons for vaping onset and continuation”. The resulting manuscript is, however, much too short to adequately address either goal.

Thank you for your comment. We have attempted to address all concerns outlined below and expanded the manuscript to provide additional information and clarifying detail where requested.

2. The abstract also states that sociodemographic and vaping characteristics will be compared with the UK general population. At no point do the authors actually explain how they are doing this comparison – they cite data from the UK census, but exactly how such data are compared with their own estimates is unknown. The authors conclude that “compared to the general population, the population who had ever vaped...” but this conclusion is not justified using the current data provided.

Thank you for your comment. We have added to the methods section on page 6 to clarify that the online survey collected sociodemographic data (age, sex, ethnicity etc.) based on questions identical to the UK census. “The survey used sociodemographic questions identical to those within the UK census...” and additionally expanded the methods on page 6 to clarify that sociodemographic characteristics in the UK general population were compared to characteristics of individuals within the survey sample. Continuous variables were compared using an unpaired t-test, and categorical variables using Fisher’s exact test. We have added to and amended the text on page 6 of the methods to specifically clarify “Additionally, study sample sociodemographic characteristics were compared with the UK general population using publicly available data from the UK census. (14) Continuous variables were compared using an unpaired t-test, and categorical variables using Fisher’s exact test” We agree with the reviewer that the wording in the abstract is misleading as it currently states “sociodemographic and vaping characteristics” were compared between the UK general population and those vaping non-nicotine drugs - when in fact this specific element of the study only compares sociodemographic characteristics between these samples, as such we have removed the phrase “and vaping” from the abstract for clarity.

We do also currently report the results of the comparison with UK census data within the main text on page 12 “When comparing the study participants who had ever vaped a non-nicotine drug to both the UK general population and to those study participants who had never vaped, they were significantly different in every studied protected characteristic sociodemographic category. This group

was significantly younger (with significantly more people aged <35 and significantly fewer aged >45), had a higher proportion of males, there were fewer people identifying with white ethnicities, there were more individuals who identified as LGBT+, more non-religious individuals, more people with disabilities, and fewer married people. However, there was no statistically significant difference by level of educational attainment. A breakdown by age, sex and ethnicity can be found in table 1, with the remaining sociodemographic differences available in the online supplementary material as table S3”, so do feel the conclusion of the abstract is justified using the data presented in the study. We have also added a column to table 1 to report the differences in sociodemographic status between the UK general population and the study sample of those who have ever vaped a non-nicotine substance.

3. The review of literature seems very inadequate. The authors state their study is the largest cross-sectional sample to examine these issues, and the only one to investigate sociodemographic associations. A growing body of research has investigated non-nicotine vaping, as well as reasons for vaping, and the authors have not recognized this body of work (see, for example, US studies by Keyes et al 2022; Miech et al 2019; Patrick et al., 2020; and UK studies by Sund et al

Thank you for your comment. We do state our introduction on page 5 that “Monitoring of nicotine vaping in the United Kingdom is well documented, (8) whilst data on vaping of non-nicotine drugs is sparse (9)”. Of the additional UK based references provided by the reviewer, one (Sund et al.) was published whilst our paper was undergoing peer-review in this journal, its senior author is Blundell whom the reviewer notes we have cited, and it only reports on nicotine and cannabis products (not on any non-nicotine psychoactive drugs). The second (Fatar et al.) reports only on England, only on cannabis and nicotine products, and does not comment on reasons for vaping non-nicotine products. We have added in the Sund and Fatar references to the introduction on page 5 but do not think the addition of these references contradict our assertion that there remains sparse data in the UK, nor does it contradict our statement that “Although some studies have examined vaping in the context of nicotine and cannabis administration, (9-11) to our knowledge, only a single cross-sectional study conducted in 2017 in adults in the United Kingdom (UK) reported that 340 participants (13.6%) had ever used an electronic vaping device for any type of recreational drug administration during their lifetime” The other references mentioned by the reviewer pertain to adolescent samples, not adult samples - which is the focus of this study, however we have included the suggested Keyes study as a reference in the introduction to highlight the current research in this area.

4. A great deal more information is needed regarding methods. What is the Prolific platform, and what are the strengths/limitations of its use? Why were data collected only on one day (March 10, 2022)? What hours? What were the limitations of this? How might it have affected results? The actual wording for measures used needs to be provided, not merely briefly described. How were missing data handled, and what was the extent of missing? Clarification of the analyses is also needed; were all models bivariate?

Thank you for your comments. We have added to the methods on page 6 to expand on what the Prolific platform is “...through the online Prolific platform. (13) Prolific, using their survey panel of research respondents based in the UK, advertised the study to all adults on their platform. (13)” reference 13 directs readers to the Prolific website which contains detailed information on their online panel survey, and we have additionally added in reference 13 by Newman et al. (Newman A, Bavik YL, Mount M, Shao B. Data collection via online platforms: Challenges and recommendations for future research. *Applied Psychology*. 2021;70(3):1380-402.) a recent article describing the challenges and utility of using such online panel surveys for research.

We have clarified the reason data collection occurred on only a single day and added to the methods section on page 6. Our pre-specified sample size calculation, available as part of the pre-registered protocol and statistical analysis plan on the Open Science Framework (<https://osf.io/bdfhp/>), suggested the minimum sample size required was 3,275. This was surpassed within 24 hours and as such data collection occurred over a single twenty-four hour period. "The survey went live on March 10th 2022 with the pre-specified minimum sample size of 3,275 being reached within twenty-four hours" We additionally note in the discussion limitations of this survey design stating on page 13 "Additionally, the day prior to data collection was national 'No Smoking Day' in the UK which may have affected estimates"

The full questionnaire is available to readers either via the Open Science Framework and we have now at the request of the editor additionally submitted the full questionnaire as an online supplementary file should readers wish to see the exact wording of any measures. We have amended text in the methods on page 6 to clarify that these two sources of the original questionnaire are available "The study protocol, questionnaire and statistical analysis plan were pre-registered on the Open Science Framework (OSF) and are available at <https://osf.io/bdfhp/>." "The final published questionnaire is additionally available in the online supplementary material"

We have added in a statement to the methods on page 6 to clarify how the survey design enabled no missing data to be generated "Participants were required only to complete sections of the applicable to their self-described vaping status of each substance and answers forced prior to moving onto subsequent section thus no missing data could be generated"

We have added in text to the methods on page 7 to clarify "...all comparisons were bivariate"

5. The analytic approach is lacking. Use of global chi square tests and t-tests does not provide the ability to identify where specific differences are found.

Thank you for your comment. The comparison of sociodemographic and vaping characteristics between two studied groups comparing proportions of individuals with specific characteristics or means for continuous means is appropriate. The same approach is used in previous cross-sectional studies within health research and in this topic area (e.g., Blundell et al.) We have added text to the methods to clarify that, where categorical variables contain more than two levels, adjusted residuals were calculated to identify cells contributing to the differences between groups, with adjusted residuals greater than 2.58 - corresponding to a significance level of $\alpha = 0.01$. We have added text to the methods on page 6 to clarify "Where there were more than two levels to a categorical variable, adjusted residuals, with a significance level of $\alpha = 0.01$, were calculated to identify which individual cells contributed to between group differences" We have additionally expanded the text in the results section on page 12 to state the specific contributing cells for the four level-age variable "This group was significantly younger (with significantly more people aged <35 and significantly fewer aged >45" This was clarified in the table one footnote but we accept the font may have been too small and have now made this more easily visible.

We have also expanded text in the discussion section page 11 to state "The small numbers of people vaping specific substances prevent robust statistical analyses due to small cell sizes and as such statistical testing should be interpreted with caution. Future studies with larger sample sizes or meta-analyses may be able to overcome these limitations"

6. The authors are to be commended on their careful reporting in the tables, but these indicate that the sample sizes used are simply too small to provide reliable estimates for many groups. Cell sizes are frequently shown of less than 5.

Thank you for your comment. We agree with the reviewer however still feel it is an important finding to demonstrate the very low overall numbers of people within our sample who are vaping specific non-nicotine substances in table 2. For the sociodemographic analysis in table 1, we used fewer levels than are shown in table 1 which avoided comparisons of small groups (except for those who did not want to disclose their sex). This was clarified in the footnote but we accept the font may have been too small and have made this more easily visible. We have additionally expanded the text in the results section on page 12 to state the specific contributing cells for the four level-age variable "This

group was significantly younger (with significantly more people aged <35 and significantly fewer aged >45”

We have nevertheless also expanded the text on the discussion section on page 11 to further highlight this limitation “The small numbers of people vaping specific substances prevent robust statistical analyses due to small cell sizes and as such statistical testing should be interpreted with caution. Future studies with larger sample sizes or meta-analyses may be able to overcome these limitations”

7. No descriptive information is provided for the overall sample.

Thank you for your comment. Descriptive statistics for the overall sample are provided in table 1 (age, sex and ethnicity) and in table S3 of the online supplementary material (Gender identity, Sexual orientation, Religion, Marital status, Disability, Level of education). Whilst we reported a breakdown of the vaping characteristics in the results section in the text on page 8, we agree there is currently no description of the sociodemographic profile of the overall sample in the text and have thus added text to the results section on page 8 to briefly describe the sociodemographic profile of the overall sample “The majority of the overall sample were aged between 25 and 44 (57.9%), predominantly identified as female (64.9%), and predominantly identified as being from a white ethnicity (80.8%)”

Minor comments

Abstract

P. 4 line 8: recommend clarifying the meaning of non-nicotine drugs, as there are flavorings, etc.

Thank you for your comment. We have added the word “psychoactive” to the abstract on page 3 to clarify that non-nicotine drugs refers to psychoactive compounds, not flavourings etc. “Electronic vaping devices are being used to consume nicotine and non-nicotine psychoactive drugs”

P. 4 line 16: provide age range

Thank you for your comment. We have added the age range of participants to the abstract on page 3 as requested. “A convenience sample of adults (aged ≥ 18 years) in the UK”

Introduction

P. 6, lines 9-10: use “confirm” and not “suggest”; this research is clear on this issue. Also, define non-nicotine drugs at first use

Thank you for your comment. We have amended the word “suggest” to “confirm” as suggested by the reviewer and have added the word “psychoactive” to the introduction on page 4 to clarify that non-nicotine drugs refers to psychoactive compounds, not flavourings etc. “...recent reports also confirm use of electronic vaping devices to administer non-nicotine psychoactive drugs”